# Nuclear Transfer Arrest Embryos Show Massive Dysregulation of Genes Involved in Transcription Pathways

**DOI:** 10.3390/ijms22158187

**Published:** 2021-07-30

**Authors:** Chunshen Long, Hanshuang Li, Xinru Li, Wuritu Yang, Yongchun Zuo

**Affiliations:** State Key Laboratory of Reproductive Regulation and Breeding of Grassland Livestock, College of Life Sciences, Inner Mongolia University, Hohhot 010020, China; cslong@mail.imu.edu.cn (C.L.); lhshuang@mail.imu.edu.cn (H.L.); 31908018@mail.imu.edu.cn (X.L.)

**Keywords:** SCNT embryos, transcription pathways, gene regulatory networks, abnormal gene expression, molecular barriers

## Abstract

Somatic cell nuclear transfer (SCNT) technology can reprogram terminally differentiated cell nuclei into a totipotent state. However, the underlying molecular barriers of SCNT embryo development remain incompletely elucidated. Here, we observed that transcription-related pathways were incompletely activated in nuclear transfer arrest (NTA) embryos compared to normal SCNT embryos and in vivo fertilized (WT) embryos, which hinders the development of SCNT embryos. We further revealed the transcription pathway associated gene regulatory networks (GRNs) and found the aberrant transcription pathways can lead to the massive dysregulation of genes in NTA embryos. The predicted target genes of transcription pathways contain a series of crucial factors in WT embryos, which play an important role in catabolic process, pluripotency regulation, epigenetic modification and signal transduction. In NTA embryos, however, these genes were varying degrees of inhibition and show a defect in synergy. Overall, our research found that the incomplete activation of transcription pathways is another potential molecular barrier for SCNT embryos besides the incomplete reprogramming of epigenetic modifications, broadening the understanding of molecular mechanism of SCNT embryonic development.

## 1. Introduction

Somatic cell nuclear transfer (SCNT) technology can reprogram terminally differentiated cell nuclei into a totipotent state to realize the cloning of animals [1]. SCNT has great prospects in therapeutic cloning, animal breeding and endangered species protection [2,3,4,5]. At present, there are still many technical obstacles in SCNT that cause SCNT embryos to have low cloning efficiency, extra-embryonic tissues and some abnormal phenomena after the birth of cloned animals [6,7]. In mice, 70% of SCNT embryos are arrested at early cleavage stages, especially from the one-cell to the two-cell stage [8,9], which greatly limits the application of SCNT technology.

In recent years, the development of low-input sequencing technology has enabled more accurate analysis of transcriptome and epigenetic dynamics during SCNT reprogramming at single-cell resolution, providing new clues for revealing and overcoming molecular defects in somatic reprogramming [9,10,11]. Many studies have shown that there are a large number of abnormal expression genes in SCNT embryos. For example, Matoba et al. found 3775 differentially expressed genes (DEGs) at the two-cell stage between in vitro fertilization (IVF) and SCNT embryos of mice [10]. Liu et al. found 6948 DEGs at the eight-cell stage between IVF and SCNT embryos of bovine [12]. At the same time, most of these abnormally expressed genes have different epigenetic characteristics from normal developmental embryos, and the ectopic expression of the corresponding epigenetic modifiers can restore the global transcriptome and improve SCNT embryonic development [9,12,13]. However, the ectopic expression of the corresponding epigenetic modifiers cannot fully rescue the abnormally expressed genes, which indicates that more factors hindering the further development of SCNT embryos need to be clarified.

In mice, zygotic genome activation (ZGA) mainly takes place in the two-cell stage embryos. During this progress, about 90% of the maternally deposited mRNAs have a degradation [14,15] in which some of the basal transcription factors (TFs) or their corresponding maternal mRNAs contribute to the activation of embryo transcription [16,17,18]. A recent analysis indicates that inhibition of minor ZGA impairs the RNA polymerase II (Pol II) pre-configuration and embryonic development in mouse embryos [19]. In accordance with our previous study, we observed that the transcripts related to transcription, such as TFIID subunits, RNA polymerase and mediators, are the main trigger genes, which are not fully activated in interspecies SCNT (inter-SCNT) embryos [20,21]. The initiation of transcription as a key point plays important roles in the regulation of gene activity during mammalian development [22]. However, whether transcription-related pathways also play roles in SCNT embryos needs to be further investigated [23].

In our study, we found incomplete activation of transcription pathways in SCNT embryos and revealed that abnormal transcription processes may impede the expression of key genes, leading to GRN defects and further affecting the crucial biological processes of embryonic development in SCNT embryos. Our study showed incomplete activation of transcription pathways functions as a barrier for SCNT embryos, which provided a theoretical basis for in-depth understanding of SCNT embryo development and improving the efficiency of nuclear transfer.

## 2. Results

### 2.1. Incomplete Activation of Transcription Pathways in SCNT Embryos

Transcription is one of the most fundamental cellular events and the first occurrence of this process is accompanied by the zygotic genome activation (ZGA) [24]. However, the potential influence of transcription-related pathways on embryo development remains elusive. To address this, we collected data on three pathways related to transcription process from the Kyoto Encyclopedia of Genes and Genomes (KEGG) database according to the functional classification information of the pathway. These three KEGG pathways are basal transcription factors (TFs) (mmu03022), RNA polymerase (mmu03020) and spliceosome (mmu03040), involving 44, 31 and 136 factors, respectively. In order to explore the developmental defects of SCNT embryos from the perspective of transcription-related pathways activation, we compared the gene expression patterns of basal TFs, RNA polymerase and spliceosome between in vivo fertilized embryos (WT) and SCNT embryos (Figure 1A). A major activation wave of the three pathways was observed at the two-cell to four-cell stage in WT embryos. However, the three transcription pathways were incompletely activated in SCNT embryos. Moreover, in nuclear transfer two- and four-cell-stage (NTA2 and NTA4) embryos, the expression levels of genes involved in basal TFs, RNA polymerase and spliceosome are significantly lower than corresponding development stages of nuclear transfer to blastocyst embryos (NTB) and WT embryos (Figure 1B).

To further explore the potential effects of aberrant transcription process on NTA embryos, we compared the activation levels of transcription pathways between NTA and NTB/WT embryos. The results indicated the heterogeneity of gene recovery in NT 2-cell to blastocyst (NTB2) and NT 4-cell to blastocyst (NTB4) embryos, in which only about half of the genes were rescued (Figure 1C). In NTB2 embryos, 20.1% of transcription associated genes were highly rescued and 29.4% of transcription associated genes were partially rescued. In NTB4 embryos, 39.7% transcription associated genes were highly rescued and 6.9% of transcription associated genes were partially rescued. At the same time, the rescued genes were distributed in all of the three pathways. In NTB2 embryos, 23, 15 and 63 genes were rescued in the basal TFs, RNA polymerase and spliceosome pathways, respectively. In NTB4 embryos, 20, 14 and 61 genes were rescued, respectively. The ratio of highly rescued genes in NTB4 embryos was greater than that in NTB2 embryos (Figure 1D,E). These results implied that the incomplete activation of transcription pathways may be an obstacle to the development of SCNT embryos.

### 2.2. Abnormal Transcription Pathway was Related to Massive Dysregulation of Genes in NTA Embryos

RNA polymerases, basal TFs and spliceosome are required for the expression of genes in the eukaryotic cell [25,26]. In recent years, many studies have shown that there are a large number of abnormally expressed genes in SCNT embryos, which is usually explained as the result of incomplete reprogramming of epigenetics modification in SCNT embryos [1,8,27,28,29,30,31]. However, in addition to epigenetic barriers, there are many potential molecular barriers that hinder the development of SCNT embryos that need to be further unraveled [9,12,13]. In our research, we found that the abnormal transcription process was significantly related to the development arrest of SCNT embryos. Then, we wondered whether the aberrant transcription pathways can lead to the dysregulation of gene expression and developmental arrest of SCNT embryos.

To this end, we detected the downstream gene regulatory networks (GRNs) between WT embryos and NTA embryos based on pySCENIC (Appendix A). pySCENIC is an algorithm that can reconstruct GRNs with transcription factors (TFs) as the core based on co-expressed and TF binding motifs analysis (see Section 4, Methods). Compared with NTA2 embryos, more predicted target genes have been observed in the GRNs of WT2 embryos and only 1067 genes shared in the two types of embryos (Figure 2A). The consistent results were also observed in WT4 embryos (Figure 2A). Notably, predicted target genes showed heterogeneous expression between WT and NTA embryos. In the 3514 WT2 embryos specific target genes, 1130 expressed more than twice as much as NTA2 embryos (Figure 2B). In 4672 WT4 embryos specific target genes, 1594 expressed more than twice as much as NTA4 embryos (Figure 2C). The GO term enrichments showed that 1130 down-regulated genes of NTA2 embryos were mainly involved in catabolic process, signal transduction, translation, histone modification, etc. (Figure 2D). In addition, 1594 down-regulated genes of NTA4 embryos were mainly enriched in regulation of interferon production, catabolic process, signal transduction, translation, etc. (Figure 2E). These findings confirmed that the aberrant transcription pathways may lead to the massive dysregulation of genes and biological process in NTA embryos.

### 2.3. Defective Activation of Transcription Pathways Downstream GRNs in NTA Embryos

To further evaluate the potential role of transcription pathway downstream GRNs on embryo development, we next sought to identify core TFs from the GRNs. The top 1 or 2 core factors with the largest targets number in three transcription pathways of WT embryos were screened, respectively, including Gtf2a2, Taf9 (basal TFs), Polr3a, Polr3g (RNA polymerase) and Ncbp1 (spliceosome) (Figure 3A). As expected, the expression levels of the five core TFs in WT embryos were higher than that in NTA embryos (except for Gtf2a2 in NTA4 embryos) (Figure 3B).

Next, we utilized the TF regulon activity obtained by pySCENIC to detect the predicted target genes of this five core TFs. Two criteria were used to identify the TFs: first, we only kept co-expressed TFs with positive correlations, i.e., potential activation associations; second, we only kept TFs whose binding motif was over-represented in the search space around the transcription start site (TSS) of genes. A specific GRN was observed in WT2 embryos, which contains the five core TFs and 951 downstream genes. No corresponding regulatory relationships have been observed in NTA2 embryos (Figure 3C). Among them, Hdac4, as a histone deacetylase, is co-regulated by Taf9, Polr3a and Ncbp1, suggesting the crucial epigenetic regulation defects in SCNT embryo development. Histone H3K9me3 demethylase Kdm4b was targeted by Taf9, which is consistent with previous studies that Kdm4b may function as a natural assistance for SCNT embryos to overcome the H3K9me3 barrier [9]. Moreover, DNA demethylase Tet1 [9,32], histone acetylase Kat2a and Kat6a and some pluripotency factors, such as Sox2, Taxa2r, Cbfa2t2, Id1, Zfp109, Gata6 and Igf2, were also the downstream predicted target genes of the five core TFs. We further checked the molecular function involved in this GRN. The 951 predicted target genes were mainly enriched in organelle organization, catabolic process, covalent chromatin modification, histone modification, stem cell population maintenance and so on (Figure 3D).

Furthermore, a more complex specific GRN was found in the WT4 embryos which contain 1440 predicted target genes, including some key pluripotency factors [33,34,35,36] (e.g., Sall1, Id1, Dppa5a, Kat6a) (Figure 4A). The genes involved in this GRN were mainly enriched in catabolic process, protein disassembly, lipid localization and regulation of cell growth (Figure 4B). These results indicated that the transcription pathways formed intricate regulatory relationships with a large number of key genes, thereby facilitating the progression of embryonic development. However, incompletely activated transcription pathways can cause defects of GRNs in NTA embryos and further lead to abnormalities in certain biological processes, such as organelle organization, basic metabolism, epigenetic modification and pluripotency acquisition.

### 2.4. NTA Embryos Showed Weak Coordination of Key Predicted Target Genes

The transcriptional state of a cell emerges from an underlying GRN [37]. In the above study, we found that NTA embryos have defective activation of transcription pathways downstream GRNs. However, how these downstream regulatory genes promote embryo development through synergistic effect needs further exploration. We first detected the expression patterns of the key predicted target genes between the WT2 and NTA2 embryos. The 16 key target genes—including epigenetic modifier Tet1, Kat2a and Kat6a and pluripotency factors Sox2, Taxa2r, Cbfa2t2, Id1, Zfp109, Zfp352, Gata6 and Igf2—have higher expression levels in the WT2 embryos compared to NTA2 embryos (Figure 5A).

Next, we evaluated the coordinate expression of key downstream genes of transcription pathways. Co-expression analysis revealed strong correlation with a Pearson’s correlation coefficient (PCC) of more than 0.55 between epigenetic modifications and pluripotency factors, with the exception of Gata6, Hdac4, Igf2 and Kdm4b (Figure 5B). We extracted the targeting relationship of these key genes from the regulons of pySCENIC and constructed the GRNs (Figure 5C). The results indicated that GRN in WT2 embryos had a more complex regulatory relationship than that in NTA2 embryos, and there were more co-regulatory relationships among various genes. However, NTA2 embryos show a defect in the coordination of these factors. This suggests that the synergism of epigenetic modification and pluripotency factors is essential to facilitate the normal development of SCNT embryos.

## 3. Discussion

At present, despite great advances in SCNT technology [3,29,38], it is far from achieving a perfect reprogramming approach [6,7,39]. Therefore, elucidating the barriers of reprogramming and finding effective ways to improve the efficiency of SCNT have become urgent issues [32]. Transcription is the most fundamental molecular event, which was crucial for the regulation of gene activity during ZGA of embryo development. For the initiation of transcription, RNA polymerase II (Pol II) can bind to basal transcription factors to form a pre-initiation complex (PIC) [40,41,42]. After transcription, eukaryotic mRNA precursors (pre-mRNA) were spliced into mature mRNA by spliceosome [43]. In our previous study, we found that the abnormal expression of transcription-related genes might be caused by the nuclear-cytoplasmic incompatibility between transferred nuclei and recipient cells in SCNT embryos [20]. The study further indicated that interspecies SCNT embryos only wasted the stored maternal mRNA of master regulators, but failed to activate their self-sustained pathway of RNA polymerases [20,21].

In this study, we found that transcription pathways were activated after major ZGA, thereby facilitating the expression of massive downstream key genes in WT mouse embryos. However, transcription pathways in NTA embryos were incompletely activated at 2-cell and 4-cell stages, which led to the down-regulation of these genes compared to WT embryos (Figure 5D). In addition, the incomplete activation of transcription pathways can lead to defects of core GRNs and biological processes related to embryo development, thereby hindering the development of SCNT embryos.

In conclusion, we identified incomplete activation of transcription pathways and massive dysregulation of genes related to transcription pathway in NTA embryos. Then, the GRNs indicated that crucial factors responsible for transcription play a coordinated role in epigenome erasure and pluripotency regulation during normal embryo development [44,45,46,47]. However, in NTA embryos, predicted target genes of transcription pathways were varying in degrees of inhibition and showed a defect in synergy. Overall, our study identified the molecular barriers and defective GRNs related to transcription pathways in SCNT embryos, which provides new insights into understanding the developmental blocks of SCNT embryos.

## 4. Materials and Methods

### 4.1. Dataset Collection

The single-cell RNA sequencing (RNA-seq) data of mouse pre-implantation embryo development were downloaded from Gene Expression Omnibus (GEO) database under accession number GEO: GSE113164 [48]. There are two embryonic types, namely somatic cell nuclear transfer (SCNT) embryos and in vivo fertilized (WT) embryos. Both SCNT and WT samples include zygote, 2-cell, 4-cell, 8-cell, morula and blastocyst, and each stage has three replicates. Moreover, another single-cell RNA-seq data (GSE70605) [9] were also reanalyzed in this study, which includes two types of SCNT embryos and in vivo fertilized (WT) embryos. These embryos can be divided into nuclear transfer 2-cell arrest embryos (NTA2), NT 4-cell arrest embryos (NTA4), NT 2-cell to blastocyst (NTB2) embryos, NT 4-cell to blastocyst (NTB4) embryos, in vivo fertilized 2-cell embryos (WT2) and in vivo fertilized 4-cell embryos (WT4).

### 4.2. RNA-seq Data Processing

For RNA-seq data processing, all RNA-seq data were controlled by FastQC software (http://www.bioinformatics.babraham.ac.uk/projects/fastqc/, accessed on 2 November 2020) and raw reads were trimmed based on Trimmomatic (version 0.38) [49] to remove low-quality samples. Next, filtered reads were mapped to the mouse mm9 genome with HISAT2 (version 2.1.0) [50] aligner with default parameters. Then, read counts of each gene were calculated using HTseq (version 0.11.0) [50]. Transcriptome assembly was performed using Stringtie (version 1.3.3) [50,51] and Ballgown (R package), and expression level of each gene were quantified with normalized FPKM (fragments per kilobase of exon model per million mapped reads) [10].

### 4.3. Transcription-Related Pathways Selection

Basal TFs (mmu03022), RNA polymerase (mmu03020) and spliceosome (mmu03040) related to transcription were obtained from Kyoto Encyclopedia of Genes and Genomes (KEGG) database (https://www.kegg.jp/kegg/pathway.html, accessed on 5 November 2020) according to the functional classification information of pathway [52]. These three KEGG pathways contain 44, 31 and 136 factors, respectively.

### 4.4. Differential Genes Expression Analysis

Differential expression analysis was performed by R package DEseq2 [53]. For each comparison, genes with a Benjamini and Hochberg-adjusted *p*-value (false discovery rate, FDR) < 0.05 and the absolute of Log2 (fold change) > 1 were regarded as differential expression genes (DEGs) [54].

### 4.5. Definition of Transcription Related Gene Rescue in NTB Embryos

The expression levels (FPKM) of genes involved in the three transcription pathways were calculated and normalized in WT, NTA and NTB embryos. All abnormally expressed genes in NTA embryos compared to WT embryos were divided into two categories, including rescue and rescue failure genes in NTB embryos. For rescued genes, the highly rescued genes were defined as follows:(1)$$\log_{2}\left(FPKM_{NTB}+1\right)>\left(\frac{\log_{2}\left(FPKM_{WT}+1\right)+\log_{2}\left(FPKM_{NTA}+1\right)}{2}\right)$$

The partially rescued genes were defined as follows:(2)$$\log_{2}\left(FPKM_{NTA}+1\right)<\log_{2}\left(FPKM_{NTB}+1\right)<\left(\frac{\log_{2}\left(FPKM_{WT}+1\right)+\log_{2}\left(FPKM_{NTA}+1\right)}{2}\right)$$

The remaining genes are considered to be unrecovered genes in NTB embryos.

### 4.6. Single-Cell Gene Regulatory Network Inference

The workflow of pySCENIC [37] (https://pypi.org/project/pyscenic/0.6.6/#tutorial, accessed on 25 December 2020) was used to identify the GRNs involved in transcription-related factors during embryonic development. In pySCENIC workflow, the RcisTarget [55] package determine TFs and their predicted target genes (i.e., targetomes) based on the correlation of gene expression across cells, and GRNBoost [56] identifies whether the predicted target genes have the corresponding motifs of TFs to refine targetomes. Finally, active targetomes were recognized in every single cell. The regulatory network centered on transcription-related factors was screened out and visualized by Cytoscape [57].

### 4.7. Functional Pathways Enrichment and Statistical Analysis

Gene Ontology (GO) enrichment analysis was performed based on the R package clusterProfiler (version 3.14.3) [58]. Statistical analyses were implemented with R (version 3.6.0, http://www.r-project.org, accessed on 19 December 2020). Student’s *t*-test was performed using the “t.test” function with default parameters, and *p*-values < 0.05 were considered statistically significant. Representative GO terms with *p*-value < 0.05 were summarized.

### 4.8. Data Visualization

In this study, R/Bioconductor (http://www.bioconductor.org, accessed on 19 December 2020) software packages were mainly used for data visualization. For example, the Venn plot was produced by using R packet VennDiagram, and the bar plot, box plot and scatter plot were generated with the R packet ggplot2 (http://ggplot2.org/, accessed on 28 December 2020).

## Figures and Tables

**Figure 1 ijms-22-08187-f001:**
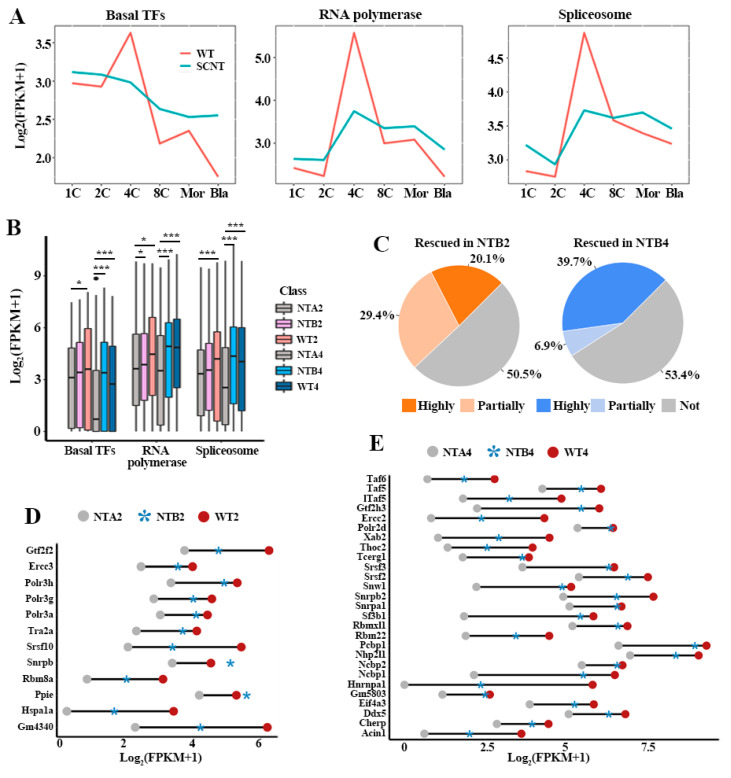
Incomplete activation of transcription-related pathways in SCNT normal and arrest embryos. (**A**) Different activation waves of three key pathways (basal TFs, RNA polymerase and spliceosome) related to transcription events was observed between WT and SCNT embryos. The expression patterns were determined by the dynamic changes in the average normalized FPKM at each developmental stage. The calculation program was conducted by the function “mean” of R. (**B**) The boxplot shows the differential activation of three pathway among NTA, NTB and WT embryos. Differences are statistically significant. (*) *p*-value < 0.05; (***) *p*-value < 0.001, *t*-test. NTA2, NT 2-cell arrest embryos; NTB2, NT 2-cell to blastocyst; WT2, in vivo fertilized 2-cell embryos; NTA4, NT 4-cell arrest embryos; NTB4, NT 4-cell to blastocyst; WT4, in vivo fertilized 4-cell embryos. (**C**) The proportion of abnormally expressed gene were rescued in NTB embryos. The definition of highly and partially rescued genes shown in method. (**D**,**E**) Heterogeneous rescue effect on gene expression at 2-cell and 4-cell stage of NTB embryos.

**Figure 2 ijms-22-08187-f002:**
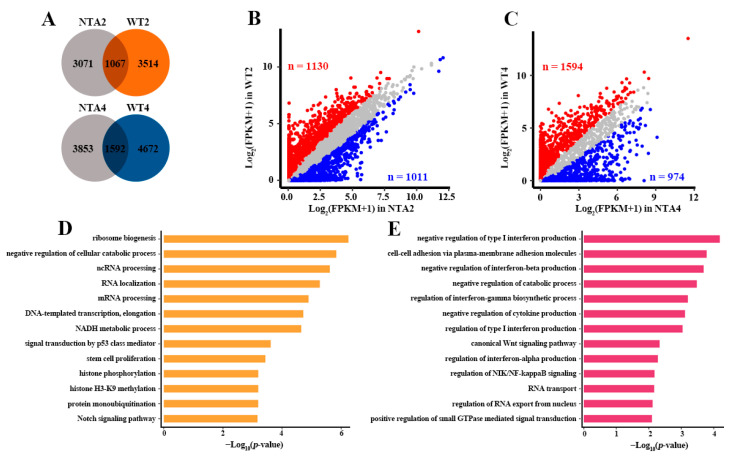
The different effects of transcription pathway on global gene expression between WT and NTA embryos. (**A**) Overlap between targets genes regulated by three pathway related TFs in NTA and WT embryos. (**B**) Expression pattern of 3514 specific predicted target genes of WT2 embryos in Figure 2A between NTA and WT embryos at 2-cell stage. The red dots represent the genes which are more than twice as expressed in WT2 as in NTA2 embryos. The blue dots represent the genes which are more than twice as expressed in NTA2 as in WT2 embryos. (**C**) Expression pattern of 4672 specific predicted target genes of WT4embryos in Figure 2A between NTA4 and WT4 embryos. (**D**) The representative GO term (biological process, BP) enrichments of WT2 up-regulated 1130 genes in Figure 2B. (**E**) The representative GO term (BP) enrichments of WT4 up-regulated 1594 genes in Figure 2C.

**Figure 3 ijms-22-08187-f003:**
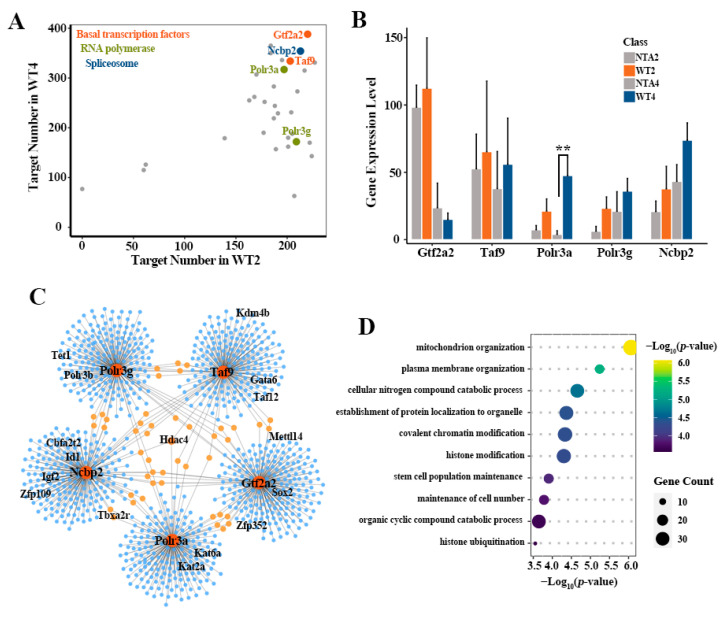
Identification of core GRN barriers related to transcription in NTA embryos. (**A**) The selection of candidate TFs (marked in figure) involved in three pathways, the five TFs (Gtf2a2, Ncbp2, Taf9, Polr3a, Polr3g) having top targets both in WT2 and WT4 embryos were screened in three pathways, respectively. (**B**) The differential expression patterns of five representative core TFs between in NTA and WT embryos. The expression patterns of these genes are represented as the average plus standard deviation (SD) of biological replicates (mean + SD). (**) *p*-value < 0.01, *t*-test. (**C**) The 2-cell stage specific gene regulatory network centered on these five TFs in WT embryos relative to NTA embryos. (**D**) The enriched biological processes for 951 predicted target genes in the GRN.

**Figure 4 ijms-22-08187-f004:**
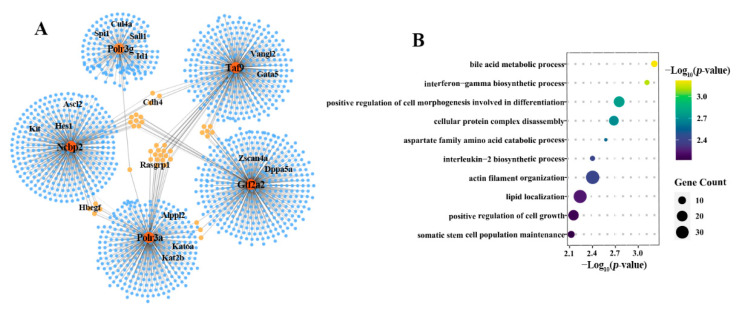
The WT4 embryo-specific GRN with five genes in the transcription pathway as the core. (**A**) The 4-cell stage specific gene regulatory network centered on the five TFs in WT embryos relative to NTA embryos. (**B**) The enriched biological processes for 1440 predicted target genes in the GRN.

**Figure 5 ijms-22-08187-f005:**
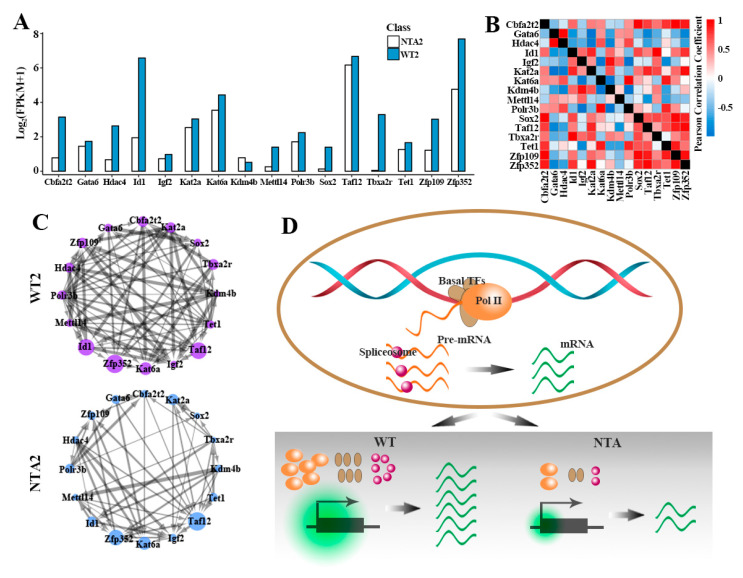
Key predicted target genes of the five TFs were dysregulated in 2-cell stage of NTA embryos. (**A**) The expression levels of the targeted pluripotency and epigenetic modification genes of the five TFs in Figure 3C. (**B**) Co-expression measured by Pearson Correlation Coefficient (PCC) clustering of genes in Figure 5A. (**C**) The gene regulated networks for the targeted pluripotency and epigenetic modification factors of the five TFs in Figure 3C in NTA2 and WT2 embryos, respectively. The dot size represents the gene expression level, and the connection line indicates targeted effect. (**D**) RNA polymerase II (Pol II) binds to basal transcription factors to form a pre-initiation complex (PIC) that turns on the initial transcription of DNA. After transcription, eukaryotic mRNA precursors (pre-mRNA) were spliced into mature mRNA by spliceosome. In WT mouse embryos, transcription pathways were activated after major zygotic genome activation (ZGA), thereby facilitating the expression of massive downstream key genes. However, transcription pathways of NTA embryos were incompletely activated at 2-cell and 4-cell stages, which led to the down-regulation of these genes compared to WT embryos. The size of green cloud indicated the transcription activation.

## Data Availability

All single-cell RNA sequencing (RNA-seq) data of mouse pre-implantation embryo development reanalyzed in this study were downloaded from Gene Expression Omnibus (GEO) database under accession number GEO: GSE113164 (https://www.ncbi.nlm.nih.gov/geo/query/acc.cgi?acc=GSE113164, accessed on 12 October 2020) [48] and GSE70605 (https://www.ncbi.nlm.nih.gov/geo/query/acc.cgi?acc=GSE70605, accessed on 12 October 2020) [9].

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
