# Peer review of "Nuclear Transfer Arrest Embryos Show Massive Dysregulation of Genes Involved in Transcription Pathways"

_ijms, 2021, doi:10.3390/ijms22158187_

Round 1
Reviewer 1 Report
In this study, the authors described that developmentally arrested nuclear transferred embryos had aberrant expression patterns associated with transcriptional pathways. While this information provides valuable information for readers in the same field, there are several points to be corrected.
(1) The authors selected basal transcription factors, RNA polymerase, and spliceosome from the KEGG pathway for analysis. The authors should provide detailed information on the reasons for the selection, such as BRITE hierarchy or IDs.
(2) The calculation program for the expression pattern in Figure 1A should be described in detail.
(3) In line 103, "the rescued genes were distributed in all of the three pathways." Specific results should be shown.
(4) In line 134, "the more predicted target genes have been observed in the GRNs of WT2 embryos and only a few genes shared in the two types of embryos." However, this is a little misleading. It is difficult to state "a few genes."
(5) Line 209, the sentence of "histone demethylase Kdm4d..." should be deleted because H3K9me3 is a histone mark inherited from donor cells and is not correlated to the expression level of Kdm4d in the embryos.
(6) Figures 1E, 2D, 2E, and 4B are not listed in the text. And the legend for Figure 1E is missing.
Author Response
Response to Reviewer 1 Comments
In this study, the authors described that developmentally arrested nuclear transferred embryos had aberrant expression patterns associated with transcriptional pathways. While this information provides valuable information for readers in the same field, there are several points to be corrected.
Point 1: The authors selected basal transcription factors, RNA polymerase, and spliceosome from the KEGG pathway for analysis. The authors should provide detailed information on the reasons for the selection, such as BRITE hierarchy or IDs.
Response 1: Thank you for your professional comments. Based on your suggestions, we have added the detailed information on the reasons for the selection in revision manuscript. The detail description is as follows: “To explore the potential influence of transcription-related pathways on embryos development, we collected three pathways related to transcription process from the KEGG database according to the functional classification information of pathway. These three KEGG pathways are Basal transcription factors (TFs) (mmu03022), RNA polymerase (mmu03020) and Spliceosome (mmu03040), involving 44, 31, and 136 factors respectively” (See Lines 77-82).
Point 2: The calculation program for the expression pattern in Figure 1A should be described in detail.
Response 2: Thank you for your professional comments. According to the suggestion, the description of calculation program for the expression pattern in Figure 1A have been added in revised manuscript (See Lines 97-99).
Point 3: In line 103, "the rescued genes were distributed in all of the three pathways." Specific results should be shown.
Response 3: Thank you for your professional comments. Based on your suggestions, specific results of rescue genes in three pathways have been added in revised manuscript (See Lines 117-119).
Point 4: In line 134, "the more predicted target genes have been observed in the GRNs of WT2 embryos and only a few genes shared in the two types of embryos." However, this is a little misleading. It is difficult to state "a few genes."
Response 4: Thank you for your professional comments. According to the suggestion, "a few genes" has been replaced by "only 1067 genes " in the revision manuscript (See Line 156).
Point 5: Line 209, the sentence of "histone demethylase Kdm4d..." should be deleted because H3K9me3 is a histone mark inherited from donor cells and is not correlated to the expression level of Kdm4d in the embryos.
Response 5: Thank you for your professional comments. According to the suggestion, the sentence of "histone demethylase Kdm4d..." has been deleted in the revised manuscript.
Point 6: Figures 1E, 2D, 2E, and 4B are not listed in the text. And the legend for Figure 1E is missing.
Response 6: Thank you for your professional comments. Based on your suggestions, Figures 1E, 2D, 2E, and 4B have been listed in the revised text (See Lines 120, 163, 165, 216). And the legend of Figure 1E has been added in the revised manuscript (See Line 106-107).
Reviewer 2 Report
In this manuscript “Nuclear transfer arrest embryos show massive dysregulation of 2 genes involved in transcription pathways”, the authors used publicly available single-cell RNA-seq data of mouse pre-implantation embryos to explore gene expression profiles between cell of SCNT embryos that failed to develop and control fertilized embryos.
Based on their findings, the authors proposed that incomplete activation of transcription pathways functions as a barrier for SCNT embryos.
Although the authors reported significant differences in gene expression profiles between the experimental groups, the significance of the findings is highly questionable. It is not clear why the authors focused their analyses comparing cells of SCNT arrested embryos with control cells from in vivo fertilized embryos. By using cells of embryo that did not develop, how can the authors make sure that the observed differences were the cause and not a consequence of the cell arrest. In other words, how can the authors be sure that arrested cells from control embryos (in vivo or in vitro fertilized) would not present the same profiles of SCNT arrested cells? In this regard, more meaningful results would be generated by compering cells of viable SCNT embryos with control embryos. Moreover, it is not an unexpected finding that target downstream genes of the genes that were detected to be abnormally expressed would also have their expression altered. For the purposes of better understanding cell reprogramming in SCNT embryos, it would be more relevant to investigate why the genes involved in transcription-related pathways where themselves abnormally regulated in the SCNT cells rather then their downstream pathways.
Other comments:
English grammar and syntax need corrections.
L12: What does heterogeneous inhibition means? The sentence is confusing.
L19: The word inhibition is being used in a wrong context, as if the authors were doing something to inhibit transcription pathways. The authors should use “incomplete activation of transcription pathways”, as used later in the text.
L56: What are inter-SCNT embryos?
L70: ZGA was used before but not defined. Acronyms should be explained at the first use, e.g., NTB2 and 4 in line 99.
L105: Should indicate they are referring to Figures 1D and E, since referring to rescued genes in NTB2 and NTB4.
Fig 3b. Are expression levels significantly different between the groups?
L252-254: The sentence is not clear.
Author Response
Response to Reviewer 2 Comments
In this manuscript “Nuclear transfer arrest embryos show massive dysregulation of genes involved in transcription pathways”, the authors used publicly available single-cell RNA-seq data of mouse pre-implantation embryos to explore gene expression profiles between cell of SCNT embryos that failed to develop and control fertilized embryos. Based on their findings, the authors proposed that incomplete activation of transcription pathways functions as a barrier for SCNT embryos.
Point 1: Although the authors reported significant differences in gene expression profiles between the experimental groups, the significance of the findings is highly questionable. It is not clear why the authors focused their analyses comparing cells of SCNT arrested embryos with control cells from in vivo fertilized embryos. By using cells of embryo that did not develop, how can the authors make sure that the observed differences were the cause and not a consequence of the cell arrest. In other words, how can the authors be sure that arrested cells from control embryos (in vivo or in vitro fertilized) would not present the same profiles of SCNT arrested cells? In this regard, more meaningful results would be generated by compering cells of viable SCNT embryos with control embryos.
Response 1: Thank you for your professional comments. As the reviewer remarked, comparing NTA embryos and in vivo fertilized embryos alone cannot directly make sure that the observed differences were the cause and not a consensus of the cell arrest. So SCNT embryos that were able to develop into blastocysts (NTB2 and NTB4 in original manuscript) were also used as control embryos in our study. A comparison of the global activation of transcription-related pathways between SCNT embryos with different fates (NTA and NTB) has been added to Figure 1B (See revised Figure 1B). The results showed the expression of transcription-related pathways in NTA embryos are significantly lower than NTB embryos. To further explore the relationship between incomplete activation of transcription-related pathways and SCNT embryo arrest, we compared the activation levels of transcription pathways between NTA and NTB/WT embryos. The results indicated that many transcription-related genes in NTB embryos were rescued to various degrees in NTB embryos, thereby enabling the normal development of SCNT embryos (Figure 1C-E). Based on the comparison of NTA embryos with normally developing SCNT embryos (NTB) and in vivo fertilized embryos, we identified incomplete activation of transcription pathways was an obstacle to the development of SCNT embryos. In addition, SCNT efficiency remains very low in terms of blastocyst development and the birth of full-term animals compared with normally fertilized embryos [1]. Therefore, it is appropriate to use in vivo fertilized embryos as control to reveal the molecular level defects of NTA embryos [2]. This is also the reason why we chose in vivo fertilized embryos (WT) as controls in addition to NTB embryos.
Point 2: Moreover, it is not an unexpected finding that target downstream genes of the genes that were detected to be abnormally expressed would also have their expression altered. For the purposes of better understanding cell reprogramming in SCNT embryos, it would be more relevant to investigate why the genes involved in transcription-related pathways where themselves abnormally regulated in the SCNT cells rather than their downstream pathways.
Response 2: Thank you for your professional comments. The reasons for the abnormal expression of transcription-related genes in NTA embryos mentioned by the reviewer have been discussed in detail in the discussion section of revised manuscript (See Lines 271-276). In our previous study, we have found that the abnormal expression of transcription-related genes might be caused by the nuclear-cytoplasmic incompatibility between transferred nuclei and recipient cells in SCNT embryos [3]. The further study indicated that interspecies-SCNT embryos only wasted the stored maternal mRNA of master regulators, but failed to activate their self-sustained pathway of RNA polymerases [3,4]. Therefore, in this study, we mainly focused on what abnormalities the incomplete activation of transcription pathways can cause at the molecular level and why the incomplete activation of transcription pathways can lead to the block of SCNT embryos.
Point 3: English grammar and syntax need corrections.
Response 3: Thank you for your comments. According to the suggestions, English grammar and syntax have been corrected throughout the revised manuscript.
Point 4: L12: What does heterogeneous inhibition means?
Response 4: We are sorry for the misunderstanding descriptions. The heterogeneous inhibition has been replaced by “incompletely activated” (See Line 12). We sincerely hope that the revised section in the resubmitted manuscript is satisfactory with you.
Point 5: L19: The word inhibition is being used in a wrong context, as if the authors were doing something to inhibit transcription pathways. The authors should use “incomplete activation of transcription pathways”, as used later in the text.
Response 5: Thank you for your professional comments. Based on your suggestions, the word “inhibition” has been replaced by “incomplete activation” in revised text (See Line 21).
Point 6: L56: What are inter-SCNT embryos?
Response 6: Thank you for your comments. The interspecies SCNT (inter-SCNT) is an ideal way for generating autologous embryonic stem cells (ESCs) and cloning endangered animal species. Interspecies SCNT (inter-SCNT) embryos is generated by which somatic nuclei introduced into the oocyte’s cytosol of a different species. And the definition of inter-SCNT embryos has been added in revised article (See Line 59).
Point 7: L70: ZGA was used before but not defined. Acronyms should be explained at the first use, e.g., NTB2 and 4 in line 99.
Response 7: Thank you for your comments. Based on your suggestions, we have added the definition of ZGA at the first use and the acronyms NTB2 and 4 have been explained in revised manuscript (See Line 51, 111).
Point 8: L105: Should indicate they are referring to Figures 1D and E, since referring to rescued genes in NTB2 and NTB4.
Response 8: Thank you for your professional comments. Based on your suggestions, the Figures 1D and E have been referred correctly in revised manuscript (See Line 120).
Point 9: Fig 3b. Are expression levels significantly different between the groups?
Response 9: Thank you for your professional comments. According to the suggestions, the significant difference of expression levels between the groups have been added in revised Figure 3B (See revised Figure 3B).
Point 10: L252-254: The sentence is not clear.
Response 10: Thank you for your professional comments. Based on your suggestions, the sentence (L252-254) has been clearly described in revised manuscript (See Line 281-286).
References
- Liu, W.; Liu, X.; Wang, C.; Gao, Y.; Gao, R.; Kou, X.; Zhao, Y.; Li, J.; Wu, Y.; Xiu, W.; et al. Identification of key factors conquering developmental arrest of somatic cell cloned embryos by combining embryo biopsy and single-cell sequencing. Cell discovery 2016, 2, 16010, doi:10.1038/celldisc.2016.10.
- Liu, Y.; Wu, F.; Zhang, L.; Wu, X.; Li, D.; Xin, J.; Xie, J.; Kong, F.; Wang, W.; Wu, Q.; et al. Transcriptional defects and reprogramming barriers in somatic cell nuclear reprogramming as revealed by single-embryo RNA sequencing. BMC genomics 2018, 19, 734, doi:10.1186/s12864-018-5091-1.
- Zuo, Y.; Su, G.; Cheng, L.; Liu, K.; Feng, Y.; Wei, Z.; Bai, C.; Cao, G.; Li, G. Coexpression analysis identifies nuclear reprogramming barriers of somatic cell nuclear transfer embryos. Oncotarget 2017, 8, 65847-65859, doi:10.18632/oncotarget.19504.
- Zuo, Y.; Gao, Y.; Su, G.; Bai, C.; Wei, Z.; Liu, K.; Li, Q.; Bou, S.; Li, G. Irregular transcriptome reprogramming probably causes thec developmental failure of embryos produced by interspecies somatic cell nuclear transfer between the Przewalski's gazelle and the bovine. BMC genomics 2014, 15, 1113, doi:10.1186/1471-2164-15-1113.
Round 2
Reviewer 2 Report
The manuscript has been overall improved.